# Job Demands and Negative Outcomes after the Lockdown: The Moderating Role of Stigma towards Italian Supermarket Workers

**Tiziana Ramaci** [1], **Stefano Pagliaro** [2], **Manuel Teresi** [2] and **Massimiliano Barattucci** [3,*]

1   Faculty of Human and Social Sciences, Kore University of Enna, 94100 Enna, Italy; tiziana.ramaci@unikore.it
2   Department of Neuroscience, Imaging and Clinical Sciences, University of Studies 'Gabriele d'Annunzio', 66100 Chieti-Pescara, Italy; s.pagliaro@unich.it (S.P.); manuel.teresi@unich.it (M.T.)
3   Faculty of Psychology, e-Campus University, 22060 Novedrate, CO, Italy
*   Correspondence: massimiliano.barattucci@uniecampus.it

**Abstract:** The Job Demands-Resources model hypothesises that some variables (especially personal and social resources/threats) moderate the relationship between job demands and work outcomes. Based on this model, in this study we examine the role of stigma towards customers as a moderator of the relationship between job demands and a series of work outcomes: that is, fatigue, burnout, and satisfaction. We advance that the relationships between work demands and outcomes should be influenced by the employee's perceptions regarding resources and constraint. In particular, we hypothesised that social stigma towards customers can represent a reliable moderating variable. Hypotheses were tested among 308 Italian supermarket workers in five supermarkets in the same chain, just after the end of the Italian lockdown caused by COVID-19. Results showed that stigma towards customers moderates the relationship between job demands and the consequences on the professional quality of life. The implications of these findings for the JD-R model are discussed.

**Keywords:** COVID-19; job demands; COVID-19 Stigma; work outcomes; self-esteem

## 1. Introduction

The emotional and behavioural reactions of the individuals on the front line during the COVID-19 lockdown and non-lockdown phases, as well as those of the population, have been the subject of scientific interest for the implementation of institutional and health communication processes, and for the study of the environmental determinants to be being able to control and to better manage the pandemic phase [1,2].

More specifically, the impact of social stigma towards possible COVID-19 carriers on the behaviour and conduct of healthcare and front-line workers has been the subject of numerous studies in the last year [3,4] and wide institutional interest [5–8].

Social stigma can indeed have a negative effect on people affected by the disease, as well as their own caregivers, their family, their friends and their community [9–12].

The current COVID-19 pandemic has resulted in social stigma and discriminatory behaviour against people belonging to certain ethnic groups and anyone believed to have been in contact with the virus [13–16].

Many studies have focused on the emotional impact and related stigma of COVID-19 in healthcare workers, and various tools have been created to measure the fear of COVID-19 and other effects on the behaviour and experiences of workers [17–19].

However, less attention has been given to both the psychological mechanisms underlying social stigma at work and to other exposed non-healthcare workers on the front line in the phases immediately following the total lockdown [20].

To the best of our knowledge, very little research has been done on the effects of social stigma on work outcomes and other organisational variables, or the possible underlying

psychological processes. Knowing the impact that social stigma has on organisational perception variables and outcomes is essential for companies to be able to cope with this post-lockdown (or discontinuity) phase and intervene appropriately.

Some indications were given by a study that investigated the relationship between stigma towards patients and other organisational (job demands, outcomes) and individual (self-esteem) variables, in a sample of Healthcare Workers (HCWs) during lockdown [21]. Ramaci and colleagues showed in particular that Job Demands, in a pandemic context, predict negative work outcomes, and that some of the stigma towards patients dimensions (discrimination and fear) have a significant relationship with work outcomes. In any event, no relationship between job demands and COVID-19 stigma toward clients was found, and evidence or indications regarding the underlying mechanism that led to stigma impacting on work outcomes are still to be explored.

From a theoretical point of view, there is also a substantial discrepancy in the conceptualisation of the stigma at work (especially in an active form, i.e., of workers towards customers) within organisational models that account for the effects on the well-being of workers. According to Major and O'Brien [22], stigma has an impact on the health of workers because it can be considered a sort of social demand or factor that reflects intergroup relationships at work, and should be considered within models such as the JD-R model to somehow go beyond or overcome the simple evaluation of working conditions.

Referring to the Job Demands model [23] we reasoned that stigma towards customers could represent a reliable moderator between JD and outcomes; the present research intends to test a possible model of the influence of stigma on the relationship between JD and outcomes, in frontline but non-healthcare workers. For this purpose, the JD measure, three different work outcomes, and self-esteem (as a resource) were used and integrated with the measure of stigma towards customers.

To the best of our knowledge, evidence about the relationship between COVID-19-related stigma and organisational variables, as investigated in the present study, is still scant. This is why the present study might provide useful information for organisational communication strategies and practical implications and activities for companies.

## 2. Work Outcomes Explained by the J-R Model

The emergence of social stigma due to the fear of contagion can generate negative behaviours in workers leading to negative work outcomes and perceived low-quality of professional life [24]. When referring to the relationships between work outcomes, and possible work and personal determinants, one of the most used conceptual frameworks is the Job Demands-Resources Model [25,26]. Influenced by the Demand-Control Model (DCM), developed by Karasek and colleagues [26], the Job Demands-Resources model (JD-R) [23] focused, and multifaceted measures of both job demands and job control that are relevant and applicable to today's working contexts, and attributes employee well-being to the characteristics of work environments.

Since its first formulation [27], the JD-R Model has been gaining increasing interest both in academia and in intervention programmes. The model has been applied in thousands of organisations and has inspired hundreds of empirical articles [28–30].

Referring to the JD-R model, the work environment is composed of job demands (physical, social, and organisational factors that require efforts and costs) and job resources (physical, social, and organisational factors that are functional in achieving work-related goals), depending on the specific organisational context [23]. The type of behaviour triggered by these resources would lead to advantages both for the individual and for the organisation [27].

The JD-R model [23,30,31], in line with the previous theories on work stress (Herzberg's Two Factors Theory [32]; Job Characteristics Model of Hackman and Oldham [33]; Demand-Control Model by Karasek [26]; Effort-Reward Imbalance Model by Siegrist [34]), hypothesises the development of the job strain (opposite to employee well-being) when the individual senses an imbalance between (JD demands) and (JR resources) that qualify their work.

Job demands (JD), which include high work pressure, an unfavourable physical environment and emotionally demanding interactions [23,35–38], are not necessarily negative but may still require an activation effort. Job resources (JR) refer to the psychological, social or organisational aspects of a job, functional to achieving objectives, reducing job demands or stimulating personal development.

Previous studies have supported the underlying predictions that job demands, psychological and unfavourable physical environment demands, are the main predictors of negative job strain [36,37], while job resources are predictors of work engagement [39].

## 3. Social Stigma, Fear for COVID-19 and Frontline Workers

During the COVID-19 pandemic period, a second form of hardship that affects the community is the stigma related to unknown people as possibly infected [40]. The conceptualisation of stigma has an extensive background; it is defined as a social process characterised by stereotyping, labelling, avoiding and discrimination towards specific categories or individuals [41]. The rapid spread of SarsCov2 saw a concurrent rise in feelings of fear and distrust that gradually overwhelmed entire countries around the world, with the increase in infections, restrictive measures to avoid infections and constantly updated media information [2].

In an unpredictable situation, individuals begin to protect themselves from a shared and invisible enemy and SarsCov2 caught the entire world off guard. It elicited a new form of stigmatisation in people towards each other, particularly towards anyone who had been in contact with possibly infected individuals [42,43]. In addition to trying to protect themselves from the disease itself, one of the most serious social problems is that people try at all costs to avoid being labelled as infected. Consequently, stigma leads to undermining social cohesion by often promoting risky behaviours, such as hiding any symptoms to avoid discrimination, creating a serious risk for others [44].

As a result of the pandemic, Taylor and colleagues [45] conceptualised a specific COVID-19 stress syndrome that highlights a range of interconnected symptoms in individuals living through the COVID-19 pandemic. The authors give an overview of how people's distress has become persistently present and how a set of attitudes and compulsive behaviours, with the fear of the danger of COVID-19 at their core, have elicited counter citizenship behaviours, such as excessive avoidance, panic buying, unsuccessful strategies of coping and high level of distress.

Recent literature shows how the weight of stigma has particularly impacted frontline categories (e.g., healthcare workers, protective services, educators, supermarket workers) that are essential in dealing with emergency situations and meeting the primary needs of the population [46,47]. Several studies have shown that healthcare workers are primarily infected during medical appointments. In a situation driven by the media, they are constantly praised and applauded by the entire nation, while behind the scenes they are seen as the main carriers of infection out of healthcare facilities [45,48].

The general pressure of the COVID-19 crisis together with the negative effects of stigmatisation has led workers to experience severe episodes of stress including while carrying out their work [49]. In fact, as in any face-to-face job, healthcare workers have also dealt with the stress generated by contact with patients infected with this new disease [50]. Although emergency workers are trained to deal with numerous types of distress (HIV, Mental Illness, Drug users, etc.) [51], they found themselves facing an unexpected situation with a lack of adequate training and personal protective equipment (PPE), primarily in the first few months of the pandemic [48]. The consequential risk is that an emergency situation leads frontline workers to have feelings of stigma towards their users or patients, this, in turn, produces stressful conditions under which to carry out one's work [45,52].

Using this evidence as a starting point, some studies have shown how different coping strategies (problem-oriented vs. emotion-oriented) have provided support for frontline workers, crucial in the absence of immediate social support due to lockdown measures [46,53]. Therefore, to strengthen the helping strategies for frontline workers,

several authors investigated a wide range of negative aspects that could affect the emotional (e.g., fear, anxiety, depression) [45,54], social (e.g., social exclusion, discrimination) [42] and occupational sphere of workers (e.g., performance, satisfaction, insecurity) [55,56]. As a result of extensive research, it was demonstrated that individuals who continued working during the first lockdown had to cope with higher stress levels than the rest of the population [48]. Coexistence of stigma received and felt, in the first months of the outbreak, created enormous difficulties, affecting individuals on several levels including the working dimension [46]. Many authors have classified the jobs most affected by the negative effects of distress generated by fear and discrimination, with healthcare workers and emergency workers (Firefighters, police) [47] coming at the top, followed by the jobs where individuals continued to work during the lockdown in highly populated environments (food manufacturing, agriculture workplaces, logistics, etc.) [57] or in face-to-face contact with consumers of basic goods and services (educators, supermarket cashiers, pharmacists) [58]. Therefore, in line with the literature, knowing what impact social stigma has on organisational perception variables is essential for companies to be able to cope with this post-lockdown (or discontinuity) phase and intervene appropriately.

## 4. Study Aims and Hypotheses

Recent contributions to the JD-R proposed that personal resources/threats can represent moderators between job demands impacting on professionals' quality of life, and many empirical studies supported the moderating role of personal resources (e.g., optimism, etc.) between JD and outcomes [59–62]. Personal resources/threats are aspects of the self that are generally linked to resiliency and refer to fundamental components of individual adaptability and coping abilities [63].

Since social stigma towards clients and customers in a pandemic scenario seems to represent an individual perception regarding personal threats at work, we believe that the JD-R model should integrate social stigma as a personal resource/threat, consequent to a social demand that reflects intergroup relationships at work (workers vs. customers) [22].

Considering the previous results of the research on HCWs [21] and the JD-R model [23,61], the present research intends to test a possible model of moderation of social stigma between JD (as antecedent) and outcomes, in frontline but non-healthcare workers. Based on previous research [21], professional quality of life, based on three dimensions, risk of Compassion Fatigue (CF), potential for Compassion Satisfaction (CS) and risk of Burnout (BO), were used as work-related health outcomes.

The role of stigma towards customers is a topic of great importance for companies all over the world, because the practical implications concern both the health of workers, the quality of services offered and compliance with restrictive government policies.

With this model, it is possible to provide new and useful feedback to the research questions, using ad hoc intervention programmes suitable for contexts and characteristics of work, workers and organisation, and improving employee professional quality of life.

What condition of Job-demands leads to Professional Quality of Life?

Based on the literature and on the rationale described above [61,64] we expect that higher levels of JD (physical and emotional load) will be related to the three work outcomes. In particular, it will be positively related to [Hp1a] Compassion Fatigue (CF) and [Hp1b] Burnout (BO), and negatively related to Compassion Satisfaction (CF) [Hp1c].

Does the stigma toward customers moderate the relation between perceived Job Demands (in similar environmental conditions) and work outcomes?

The literature [23,24] indicates that, in addition to personality differences (as stress-protective factors), personal resources/threats moderate the effect of JD on outcomes. Perceived stigma towards customers as COVID-19 carriers can lead to more serious direct consequences for workers' outcomes and their performance [21]. Based on the JD-R model and previous studies' results [21,61] it is possible to hypothesise that when workers experience increased stigma-related stress, the relationship between JD and outcomes would be affected [Hp2]. Since self-esteem previously showed no significant relationships

with negative work outcomes and a very weak correlation with COVID-19 stigma [21] we considered self-esteem as a covariate of social stigma.

More specifically, we expect that stigma would affect the relationship between JD and negative outcomes (fatigue, Hp2a; burnout, Hp2b), and, in the opposite direction, satisfaction, Hp2c).

## 5. Materials and Methods

### 5.1. Sample and Procedure

In Italy, both during the first lockdown (from March to April 2020) and in the immediately following partial re-opening phase of containment of the pandemic (from 18 May to 15 June 2020), supermarket workers in the food sales sector were, among non-healthcare workers, one of the groups most exposed on the front line to customers possibly carrying COVID-19.

A correlational study was then carried out from 20 May to the end of June, involving voluntarily participating supermarket workers from 5 different stores in the same chain, and from the same geographical area (central Italy).

In accordance with the Helsinki Declaration and APA ethical standards, previous to administration of the questionnaire, workers (a) were informed about their right to refuse to participate or to withdraw at any time, (b) confirmed that they fully understood the instructions, (c) were informed about all relevant aspects of the study and (d) assented in writing to take part in the study. They were also requested not to mention their name anywhere in the questionnaire to ensure anonymity. All data were managed according to the EU General Data Protection Regulation (GDPR) and the study was approved by the ethics committee of the E-Campus University (registration number 03/2020).

Overall, of the 413 total workers from the 5 involved supermarkets, 316 participated by filling out the paper questionnaire during working hours (response rate = 76%) and 8 subjects did not fully complete the questionnaire (data missing > 5%) so were excluded from statistical analysis.

The analysis sample consisted of 308 supermarket workers, equally distributed among the stores, and balanced also in relation to the internal distribution of roles, mainly assigned the duties of checkout operator or food sales assistant (N = 211, 68.5%), and to a lesser extent those of warehouse worker (N = 83, 26.9%) or director/coordinator (N = 14, 4.6%).

Workers were young adults (ranging from 20 to 62 years; mean age = 42.2 years; SD = 9.6), with a slightly higher presence of women (N = 165, 53.6%), a substantial portion of married subjects (N = 209, 67.9%), mostly with children (N = 175, 56.8%) and a high school diploma (N = 171, 55.5%).

Two thirds of the workers had permanent contracts (N = 206, 66.9%) and full-time hours (N = 257, 83.4%), while the mean work seniority was 13.3 years (SD = 9.5).

Following literature indications [65], a statistical power analysis was conducted through G*Power software to check for sample appropriateness; the minimum required sample emerged as 270 (2 predictors including the interaction effect, effect size level = 0.15, $\alpha$ = 0.05, power requirement of 0.80).

### 5.2. Measures

The following Self-administered questionnaires made up the questionnaire submitted to the workers:

Job Content Questionnaire (JCQ) [66] in the Italian adaptation form [67]. It is a self-administered instrument designed to measure the social and psychological characteristics of jobs. The 49 items are used to measure the high-demand/low control/low-support model of job strain development, to assess job strain. The demand/control model predicts, first, stress-related risk and, second, active-passive behavioural correlates of jobs. For the purposes of this study, we used the high work pressure demands (Psychological demands: 5 items; e.g., "In my job, I need to . . . " "work fast" "intense concentration"),

with items rated on a 4-point response scale, ranging from "definitely no" to "definitely yes" (Cronbach's alpha = 0.74).

Social Stigma toward Patients due to COVID-19 Scale (SSPCS) [21]. The questionnaire was adapted from the original [68] changing items to customers/clients rather than patients.

A self-administered questionnaire about attitudes of: discrimination (4 items) (e.g., "You feel it is not worth serving people who are most at risk of contracting the COVID-19 virus"); non-acceptance (4 items) (e.g., "If a colleague or one of their relatives has frequent contact or works with people who have contracted the virus, I would advise them to change department or job"); and fear towards clients (4 items) (e.g., "the best way to prevent COVID-19 infection is to avoid any contact with customers who have contracted COVID-19"). Cronbach's alpha for Stigma Discrimination was 0.84, 0.72 for Stigma non-acceptance and for 0.89 Fear.

The Professional Quality of Life Scale (ProQoL) developed by Stamm [69], Italian adaptation by Palestini and colleagues [70], aims to measure professional quality of life. Compassion Satisfaction (CS), Compassion Fatigue (CF) and Burnout (BO) are three aspects of Professional Quality of Life. Both the positive and negative aspects of doing your work influence your professional quality of life. Compassion satisfaction is about the pleasure you derive from being able to do your work well. Higher scores on the Compassion Satisfaction subscale (8 items) indicate that the respondent is experiencing higher satisfaction with their ability (e.g., "My work makes me feel satisfied"). Compassion Fatigue is about your work-related secondary exposure to stressful events. Higher scores on the Compassion Fatigue subscale (7 items) indicate that the respondent is at higher risk of compassion fatigue (e.g., "I feel bogged down by the system"). Burnout is associated with feelings of hopelessness and difficulties in dealing with work or in doing your job effectively. Higher scores on the burnout sub-scale (7 items) indicate that the individual is at risk of experiencing symptoms of burnout (e.g., "I feel worn out because of my work"). The items are rated on a 5-points scale from never to very often, and the scale can be used in many types of professions. The Cronbach's alpha coefficient Compassion Satisfaction was 0.91, 0.88 for Compassion Fatigue and 0.87 for Burnout. Items are rated on a 4-point response scale, ranging from strongly disagree to strongly agree.

Rosenberg Self-Esteem Scale [71], in the Italian version [72], is a 10-item scale that measures global self-worth by measuring both positive and negative feelings about the self (e.g., "I feel that I have a number of good qualities"). All items are answered using a 4-point Likert scale format ranging from "strongly agree" to "strongly disagree". The scale is believed to be unidimensional (Cronbach's alpha = 0.90).

*Socio-Demographic Variables*–Gender, age, education, and organisational variables such as position/role and seniority were considered.

### 5.3. Data Analysis

Since all variables were measured through a single questionnaire, the present correlational research applied the following methods to address response bias and issues related to the common method variance. Scales were graphically separated, two different versions of the questionnaire with different scales' sequence were used for data collection, and different scale formats and endpoints were present for all the measures [73].

In order to check for differences in the measured variables in relation to socio-demographical and organisational variables, the study carried out independent t-test, ANOVA, and correlational analysis, using SPSS 23.

Relationships between variables were investigated with correlation analysis and multiple regressions, using SPSS 23 and Process macro 3.3.

We ran a series of moderation analyses to verify whether and which aspects of COVID-19-related stigma moderated the relationship between job demands and our main outcomes: fatigue, burnout, and satisfaction. For each analysis, we ran process model number 1 in the macro developed by Hayes [74] and estimated the relationship between the predictor and the criterion at low (M − 1SD), medium and high (M + 1SD) levels of the supposed

moderator. In each moderation model, self-esteem was co-varied out in order to control for the personal resources that may potentially influence the reactions to job demands.

## 6. Results

No gender differences emerged for any of the measured variables, and no relationship between age or seniority and measured variables was found. The ANOVA did not reveal any significant differences between groups regarding the level of education, marital status and job profiles, for any of the considered variables.

Differences between permanent and fixed-term workers were found to be significant for Job Demands ($t_{307}$ = −2.79, $p < 0.01$; Fixed-term workers, mean = 1.93, $SD$ = 0.47; Permanent workers, mean = 2.1, $SD$ = 0.51) and non-acceptance ($t_{307}$ = −2.42, $p < 0.05$; Fixed-term workers, mean = 2.02, $SD$ = 0.75; Permanent workers, mean = 2.23, $SD$ = 0.67). T-test analysis highlighted that full-time workers have higher levels of negative outcomes compared to those part-time, both for fatigue ($t_{307}$ = 3.31, $p < 0.001$; Full-time workers, mean = 1.58, $SD$ = 0.91; Part-time workers, mean= 1.13, $SD$ = 0.72) and burnout ($t_{307}$ = 2.57, $p < 0.01$; Full-time workers, mean = 1.66, $SD$ = 0.87; Part-time workers, mean = 1.40, $SD$ = 0.65).

Table 1 presents descriptive statistics and zero-order correlations among the measured variables. As expected, job demands were positively related to Compassion Fatigue [Hp1a] and Burnout [Hp1b], whereas they were negatively related to Compassion Satisfaction [Hp1c].

**Table 1.** Descriptive statistics and zero-order correlations among measured variables.

| | M (SD) | 1 | 2 | 3 | 4 | 5 | 6 | 7 |
|---|---|---|---|---|---|---|---|---|
| 1. Job demands [1–4] | 2.05 (0.51) | - | | | | | | |
| 2. Stigma Discrimination [1–4] | 1.71 (0.67) | 0.196 ** | - | | | | | |
| 3. Stigma Fear [1–4] | 1.82 (0.58) | 0.214 ** | 0.615 *** | - | | | | |
| 4. Stigma Non-acceptance [1–4] | 2.16 (0.71) | 0.268 ** | 0.720 *** | 0.552 *** | - | | | |
| 5. Self-esteem [0–3] | 1.00 (0.92) | 0.057 | 0.009 | 0.085 | 0.056 | - | | |
| 6. Fatigue [0–5] | 1.51 (0.90) | 0.382 *** | 0.373 *** | 0.467 *** | 0.397 *** | 0.225 ** | - | |
| 7. Burnout [0–5] | 1.62 (0.84) | 0.413 *** | 0.419 *** | 0.485 *** | 0.429 *** | 0.112 * | −853 *** | - |
| 8. Satisfaction [0–5] | 3.75 (0.95) | −0.394 ** | −0.554 *** | −0.578 *** | −0.524 *** | −0.135 * | −0.693 *** | −0.721 *** |

$* p < 0.05$, $** p < 0.01$, $*** p < 0.001$.

### 6.1. Fatigue as an Outcome

A moderation analysis was conducted with job demands as predictor, fatigue as outcome and discrimination as the supposed moderator. Self-esteem emerged as a significant covariate ($b$ = 0.19, 95% CI: [0.1024, 0.2883]). The overall equation was significant, $R^2$ = 0.30, $F(4, 303)$ = 33.01, $p < 0.001$. Crucially for the present purpose, the job demands by discrimination interaction significantly increased the explained variance [$\Delta R^2$ = 0.02, $F(1, 303)$ = 10.27, $p$ = 0.001]. The relationship between job demands and fatigue was significant for medium ($b$ = 0.48, CI: [0.3061, 0.6587]) and high ($b$ = 0.66, CI: [0.4783, 0.8476]) levels of discrimination, while it was not significant for low levels of discrimination ($b$ = 0.24, CI: [−0.0206, 0.4948]). This outcome implies that the more employees perceived discrimination, the stronger the positive relationship between job demands and fatigue.

A moderation analysis was conducted with job demands as predictor, fatigue as outcome and fear as the supposed moderator. Self-esteem emerged as a significant covariate (b = 0.16, 95% CI: [0.0764, 0.2572]). The overall equation was significant, $R^2$ = 0.34, $F(4, 303)$ = 33.01, $p < 0.000$. Crucially for the present purpose, the job demands by fear interaction significantly increased the explained variance [$\Delta R^2$ = 0.01, $F(1, 303)$ = 5.97, $p$ = 0.015]. The relationship between job demands and fatigue was significant for low

(b = 0.32, CI: [0.0993, 0.5500]) medium (b = 0.42, CI: [0.2404, 0.6042]) and high (b = 0.62, CI: [0.4310, 0.8044]) levels of fear. This outcome implies that the more employees perceived fear, the stronger the positive relation between job demands and fatigue.

A moderation analysis was conducted with job demands as predictor, fatigue as outcome and non-acceptance as the supposed moderator. Self-esteem emerged as a significant covariate (b = 0.18, 95% CI: [0.0909, 0.2752]). The overall equation was significant, $R^2 = 0.31$, $F(4, 303) = 34.89$, $p < 0.000$. Crucially for the present purpose, the job demands by non-acceptance interaction significantly increased the explained variance [$\Delta R^2 = 0.04$, $F(1, 303) = 17.55$, $p = 0.000$]. The relationship between job demands and fatigue was significant for medium (b = 0.35, CI: [0.1665, 0.5426]) and high (b = 0.69, CI: [0.5041, 0.8907]) levels of discrimination, while it was not significant for low levels of discrimination (b = 0.13, CI: [−0.1247, 0.3767]). This outcome implies that the more employees perceived non-acceptance, the stronger the positive relation between job demands and fatigue.

Globally, the three dimensions of stigma towards customers (fear, discrimination and not acceptance) affect the relationship between Job demand and fatigue at different levels [Hp2a confirmed; see Table 2].

**Table 2.** Significance test of the moderating effect of Stigma toward customers on the relationship between Job Demands and outcomes.

| | Stigma Discrimination | | Stigma Fear | | Stigma Non-Acceptance | |
|---|---|---|---|---|---|---|
| | $\Delta R^2$, p | Hypoth. | $\Delta R^2$, p | Hypoth. | $\Delta R^2$, p | Hypoth |
| JD > Fatigue | $\Delta R^2 = 0.02$ *** | Supported | $\Delta R^2 = 0.01$ * | Supported | $\Delta R^2 = 0.04$ *** | Supported |
| JD > Burnout | $\Delta R^2 = 0.01$ * | Supported | $\Delta R^2 = 0.007$ ° | Not supp. | $\Delta R^2 = 0.34$ *** | Supported |
| JD > Satisfaction | $\Delta R^2 = 0.003$ | Not supp. | $\Delta R^2 = 0.002$ | Not supp. | $\Delta R^2 = 0.02$ *** | Supported |

$^\circ$ $p < 0.060$, * $p < 0.05$, *** $p < 0.001$.

### 6.2. Burnout as an Outcome

A moderation analysis was conducted with job demands as predictor, burnout as outcome and discrimination as the supposed moderator. Self-esteem did not emerge as a significant covariate (b = 0.08, 95% CI: [−0.0077, 0.1651]). The overall equation was significant, $R^2 = 0.31$, $F(4, 303) = 33.01$, $p < 0.000$. Crucially for the present purpose, the job demands by discrimination interaction significantly increased the explained variance [$\Delta R^2 = 0.01$, $F(1, 303) = 5.79$, $p = 0.016$]. The relationship between job demands and burnout was significant for low (b = 0.34, CI: [0.1040, 0.5833]) medium (b = 0.51, CI: [0.3510, 0.6789]) and high (b = 0.64, CI: [0.4693, 0.8128]) levels of fear. This outcome implies that the more employees perceived discrimination, the stronger the positive relationship between job demands and burnout.

A moderation analysis was conducted with job demands as predictor, burnout as outcome and fear as the supposed moderator. Self-esteem did not emerge as a significant covariate (b = 0.05, 95% CI: [−0.0340, 0.1348]). The overall equation was significant, $R^2 = 0.34$, $F(4, 303) = 40.13$, $p < 0.000$. Crucially for the present purpose, the job demands by fear interaction significantly approached significance ($\Delta R^2 = 0.007$, $F(1, 303) = 3.52$, $p = 0.061$). The relationship between job demands and burnout was significant for low (b = 0.39, CI: [0.1884, 0.6092]) medium (b = 0.47, CI: [0.2990, 0.6387]) and high (b = 0.61, CI: [0.4347, 0.7833]) levels of fear. This outcome implies that the more employees perceived fear, the stronger the positive relation between job demands and burnout.

A moderation analysis was conducted with job demands as predictor, burnout as outcome and non-acceptance as the supposed moderator. Self-esteem did not emerge as a significant covariate (b = 0.06, 95% CI: [−0.0209, 0.1507]). The overall equation was significant, $R^2 = 0.32$, $F(4, 303) = 35.76$, $p < 0.000$. Crucially for the present purpose, the job demands by non-acceptance interaction significantly increased the explained variance [$\Delta R^2 = 0.03$, $F(1, 303) = 12.75$, $p = 0.000$]. The relationship between job demands and

burnout was significant for medium (b = 0.39, CI: [0.2217, 0.5718]) and high (b = 0.67, CI: [0.4889, 0.8488]) levels of non-acceptance, while it was not significant for low levels of (non-) acceptance (b = 0.21, CI: [−0.0179, 0.4488]). This outcome implies that the more employees perceived discrimination, the stronger the positive relationship between job demands and burnout.

Globally, the three dimensions of stigma towards consumers (fear, discrimination and non-acceptance) affect the relationship between Job Demand and Burnout at different levels [Hp2b confirmed; see Table 2]

### 6.3. Satisfaction as an Outcome

A moderation analysis was conducted with job demands as predictor, satisfaction as outcome and discrimination as the supposed moderator. Self-esteem emerged as a significant covariate (b = 0.04, 95% CI: [−0.2075, −0.0256]). The overall equation was significant, $R^2 = 0.41$, $F(4, 303) = 52.22$, $p < 0.000$. There is no evidence of a moderating effect of discrimination on the relationship between job demand and satisfaction [$\Delta R^2 = 0.003$, $F(1, 303) = 1.71$, $p = 0.193$].

A moderation analysis was conducted with job demands as predictor, satisfaction as outcome and fear as the supposed moderator. Self-esteem did not emerge as a significant covariate (b = 0.04, 95% CI: [−0.1676, 0.0139]). The overall equation was significant, $R^2 = 0.42$, $F(4, 303) = 54.37$, $p < 0.000$. There is no evidence of a moderating effect of fear on the relationship between job demand and satisfaction [$\Delta R^2 = 0.002$, $F(1, 303) = 0.91$, $p = 0.341$].

A moderation analysis was conducted with job demands as predictor, satisfaction as outcome and non-acceptance as the supposed moderator. Self-esteem emerged as a significant covariate (b = 0.05, 95% CI: [−0.1887, −0.0013]). The overall equation was significant, $R^2 = 0.37$, $F(4, 303) = 45.09$, $p < 0.000$. Crucially for the present purpose, the job demands by non-acceptance interaction significantly increased the explained variance [$\Delta R^2 = 0.02$, $F(1, 303) = 10.06$, $p = 0.001$]. The relationship between job demands and satisfaction was significant for medium (b = −0.38, CI: [−0.5760, −0.1937]) and high (b = −0.65, CI: [−0.8451, −0.4522]) levels of non-acceptance, while it was not significant for low levels of non-acceptance (b = 0.21, CI: [−0.4638, 0.0458]). This outcome implies that the more employees perceived non-acceptance, the stronger the negative relationship between job demands and satisfaction.

Globally, Hp2c was partially confirmed [see Table 2], since when testing the moderation role of the three dimensions of stigma towards consumers (fear, discrimination and non-acceptance) on the relationship between job demands and satisfaction, only non-acceptance dimension emerged as a significant moderator.

Referring to the JD-R model [61], the fact that the results showed that moderation by stigma is confirmed for negative outcomes and less for satisfaction, could confirm the difference in the processes expressed by the model that sees JDs, on the one hand, impacting on negative outcomes, with job resources associated with positive outcomes. The stigma would seem to act as a sort of booster in this relationship as if it were able to increase the effect of the perceptions of JD on outcomes only when both JD and stigma reach high levels. More complex to justify based on the JD-R model is the role of non-acceptance as a moderator, being able to impact the relationship between JD and both negative and positive outcomes: non-acceptance could have a general impact on outcomes but future research will certainly have to confirm the present results.

## 7. Discussion

Theoretical models explaining the relationship between job demands and job outcomes [28–30] have recently been applied to the investigation of job stress among frontline workers during the COVID-19 pandemic. In this vein, we conducted a cross-sectional study just at the end of the Italian lockdown with the aim of testing the possible moderating role of COVID-19-related stigma towards customers (as personal resources/threats) on the

relationship between job demands and job outcomes. The results initially confirmed that high levels of job demands are positively related to the fatigue and burnout perceived by workers during the lockdown period; on the contrary, an intense demand for work was negatively related to job satisfaction. Over and above these correlations, moderation analyses showed that the relationship between job demands and outcomes (fatigue, burnout and satisfaction) is in fact moderated by the different facets of stigma toward customers (that is, fear, discrimination and non-acceptance). In particular, results showed that each considered dimension of stigma moderates the relationship between job demands and fatigue perceived by workers. In a similar way, considering the perception of burnout as an outcome, there is a significant moderation of discrimination and non-acceptance dimensions in the relationship, but it is weakened by fear. Finally, the relationship between JD and Satisfaction was significantly moderated by non-acceptance, but not by fear and discrimination.

Given the growing interest in the literature regarding the effects of stigma during the pandemic period [40] the study explains an under-investigated side of workers' life in an emergency situation such as the COVID-19 lockdown. Compared to the large amount of literature that considered stigma as a burden on frontline workers, the aim of this study was to show that, as for each individual during the pandemic, workers also feel a sense of fear and discrimination towards other people. In this way, stigma impacts not only on stress dimension of workers from the outside (from customers/patient/user to frontline workers) [45], but also inside work-life experience (from frontline workers to customers).

Furthermore, to the best of our knowledge, this study showed the efficacy of an innovative model to study deeply the relationship between job demands and some aspects of the personal experience of workers. In fact, JD-R theory continues to inspire researchers and practitioners who want to promote employee well-being and effective organisational functioning. The reason is to be found in the possibility of the model offering a way to carry out an in-depth analysis of the dynamics of working well-being which, on one hand, are driven by work demands and resources and by performance/discomfort indicators, and on the other, can be regarded as suitable to contexts and characteristics of work, workers and organisation. Furthermore, the JD-R Model [23,24] assumes that two processes (energetic and motivational) are activated regarding work behaviour impacting on professional quality of life in workers, and promoting healthy workplaces able to cope with challenges such as the current scenario of the recent public health emergency. On one side, high job demands (e.g., psychological overload) require employees to mobilise active coping responses that over time may deplete or expend energy leading to exhaustion [27], or may adopt a passive coping response, which is characterised by disengagement or burnout. The second process is described as a motivational process in which job resources can either be intrinsically motivating and foster employee growth and learning and leads to a higher level of engagement.

Sometimes to restore balance, it is not always necessary to work on reducing job demands but on increasing perceived personal resources and reducing stigma-related stress.

Since the results indicate that social stigma has an impact on workers' outcomes, we suppose that these factors may influence worker compliance and can guide management communication strategies relating to pandemic risk for workers and customers.

Much research to date has not dealt with worker stigma towards customers and users, and its effect on work outcomes, focusing mainly on self-perceived stigma; equally, few studies have attempted to integrate this construct within the JD-R model, proposing an interpretation of the stigma as a personal threat expression of a work-related social demand, and inserting it in the JD-R model as a moderator. In addition, the present research contributes to knowledge relating to the mechanisms of stigma in the pandemic context, in the context of workers exposed on the front line, and by extending the knowledge derived so far to non-health workers.

Of course, the study has some limitations that are worth noting and that could be considered in future work. Firstly, the nature of the data is cross-sectional. Future studies

may be directed at disentangling the causal direction between job demands and positive and negative outcomes of workers' life, and extend the range of considered outcomes (e.g., well-being, turnover, work tension). Furthermore, it would be interesting to extend the research to other frontline workers, such as firefighters, law enforcement officers and couriers, workers who maintained contact, albeit limited, with other unknown people during the pandemic. Finally, the study was carried out in close temporal proximity to the lockdown period imposed by the government to attempt to flatten the curve of the pandemic and we used a convenient sample. Given the unpredictability of the evolution of COVID-19 and future political choices, the data are still constantly updated to provide further support for the model presented in this document.

## 8. Conclusions

There is now a greater focus than ever on studying stigma in relation to healthcare workers.

Very little research has investigated both the psychological mechanisms underlying social stigma at work, and to other exposed non-healthcare workers on the front line in the phases immediately following the total lockdown.

The impact of stigma is serious. Knowing the impact that social stigma has on organisational perception variables and outcomes is essential for companies to be able to cope with this post-lockdown phase and intervene appropriately. Our findings underline that stigma moderates the relationship between job demands and outcomes: fatigue, burnout and satisfaction, among customers. Therefore, strengthening HRM for frontline providers requires measures to reduce stigma.

Since results indicated that social stigma impacts workers' outcomes, we suppose that these factors may influence worker compliance and can guide management communication strategies relating to pandemic risk for workers and customers.

HRM should continue endeavours to lessen the work pressure that is produced by anti-COVID-19 procedures and provisions. Specific training, organisational interventions, meetings and the opportunity to access counselling services, appear to be vital instruments to prevent and combat the effects of negative outcomes and social stigma.

**Author Contributions:** Conceptualization, M.B. and T.R.; methodology, M.B. and S.P.; validation, T.R. and M.T.; formal analysis, M.T., S.P. and M.B.; investigation, M.B. and T.R.; data curation, M.B. and M.T.; writing—original draft preparation, S.P. and M.B.; writing—review and editing, M.B, S.P., T.R. and M.T.; visualization, M.B.; supervision, S.P. and T.R. All authors have read and agreed to the published version of the manuscript.

**Funding:** This research received no external funding.

**Institutional Review Board Statement:** The study was conducted according to the guidelines of the Declaration of Helsinki, and approved by the Ethics Committee of the E-Campus University (registration number 03/2020).

**Informed Consent Statement:** Informed consent was obtained from all subjects involved in the study.

**Conflicts of Interest:** The authors declare no conflict of interest.

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
