# Peer review of "Job Demands and Negative Outcomes after the Lockdown: The Moderating Role of Stigma towards Italian Supermarket Workers"

_sustainability, doi:10.3390/su13137507_

Round 1
Reviewer 1 Report
Thank you for the opportunity of reading "Job demands and negative outcomes after the lockdown: the moderating role of stigma towards Italian supermarket workers"
The topic of the paper is quite interesting. The labour market is facing important changes in terms of demands and the paper try to respond to this issue.
The aim of the paper should be better clarified soon as possible in the introduction. It seems that the main aim has been recalled often while this is not necessary.
There are several repetitions of concepts that make the reader confused.
I suggest to reread the paper with the aim of semplifying it eliminating repetitions.
I could be useful to have a reading at Nappo BMC Public Health (2020) 20:1143 "Job stress and interpersonal relationships
cross country evidence from the EU15: a
correlation analysis" which focuses on the importance of interpersonal relationships on the job in the framework of the "job strain model".
It would help the reading a better explanation of the methodology used and the insertion of a table with the main results.
A good revision of the English is needed.
Author Response
There are several repetitions of concepts that make the reader confused. I suggest to reread the paper with the aim of simplifying it eliminating repetitions. |
Thanks for this suggestion; we carefully checked the paper and found those repetitions; we re-phrased the contents (lines 39, 60, 80] |
I could be useful to have a reading at Nappo BMC Public Health (2020) 20:1143 "Job stress and interpersonal relationships cross country evidence from the EU15: a correlation analysis" which focuses on the importance of interpersonal relationships on the job in the framework of the "job strain model". |
Thank you for your useful suggestion; we added at line 78 the suggested ref. [Nappo N. Job stress and interpersonal relationships cross country evidence from the EU15: a correlation analysis. BMC Public Health. 2020 Jul 20;20(1):1143. doi: 10.1186/s12889-020-09253-9. PMID: 32689996; PMCID: PMC7372642.] |
It would help the reading a better explanation of the methodology used and the insertion of a table with the main results. |
Thanks for this suggestion, we have clarified some methodological aspect, as also other reviewer suggested. Main results are in table 2. |
A good revision of the English is needed. |
The manuscript has been proofread by a professional native-speaker. Please find attached the certificate of professional proofreading. |

Reviewer 2 Report
The study investigated the relationships between job demands, stigma (discrimination, fear, and non-acceptance), fatigue, burnout, and satisfaction. My comments are as follows:
- The introduction should be strengthened on why the variables were chosen for the study model. I understand why they were chosen from a practical perspective but from an academic viewpoint, I think it needs to be further explained.
- There are similar studies without the COVID context in previous literature (e.g., Schaufeli & Bakker, 2004). What makes this study really distinct? I would assume it can be the moderating variable.
- The hypothesis development needs to be strengthened with theory.
- As there are multiple moderators, I do not recommend using the Process macro. I am assuming the author(s) used Model 1. When there are multiple moderators, it is recommended to analyze them all together as individuals may not be really able to perceive the moderators exclusively on their own.
- According to See et al.’s (2011) article, stigma is a multi-dimensional construct that was analyzed with first order and second order factor analyses. I would recommend multi-dimensional variables to be analyzed accordingly as well. The results may be easier to explain, especially as the current analysis form has significant and insignificant moderation findings. Another idea could be to test for a 4-way interaction.
- Was stigma measured by a Likert scale? The manuscript states it was a multiple choice measure.
- What is the purpose of self-esteem for the study?
- Regression tables should be included.
- The findings need to be further explained, especially with the moderation results.
Author Response
The introduction should be strengthened on why the variables were chosen for the study model. I understand why they were chosen from a practical perspective but from an academic viewpoint, I think it needs to be further explained. |
The choice of variables has been extensively commented on in the introduction and is focused on both the JD-R model and our previous theoretical model in which the measured variables were already present. In the literature review, we have still amply justified the choice of variables To better clarify this, we introduced a sentence at the end of the introduction (line 63)
“For this purpose, referring to the JD-R model, the JD measures, 3 different work outcomes, and self-esteem (as a resource) were used and integrated with the measure of stigma towards customers.” |
There are similar studies without the COVID context in previous literature (e.g., Schaufeli & Bakker, 2004). What makes this study really distinct? I would assume it can be the moderating variable. |
Yes, you’re right. It’s like this. What sets our study apart is the moderator variables and the pandemic context
|
The hypothesis development needs to be strengthened with theory. |
The development of the hypotheses occupies lines 168-207. The first part (lines 168-189) summarizes the JD-R model providing preparatory elements for the integration within it of the variable stigma. There are essentially two hypotheses: (1) the confirmation that the JDs impact the outcomes (with broad theoretical references also provided previously); (2) the stigma moderation testing, where the theoretical elements justifying possible moderation are provided; your suggestion allowed us to carry out a more in-depth bibliographic search on the subject of JD-R and stigma, and we were able to find another article that attempted to integrate stigma as a moderator into the JD-R model. We have consequently included this article as a reference to provide more support for the theoretical justification of the hypotheses (expecially for the Hp2). Thank you for this opportunity. Barbier, M.; Dardenne, B.; Hansez, I. A longitudinal test of the Job Demands – Resources model using perceived stigma and social identity, Eur. J. Work Organ. Psychol. 2013, 22(5), 532-546 |
As there are multiple moderators, I do not recommend using the Process macro. I am assuming the author(s) used Model 1.
When there are multiple moderators, it is recommended to analyze them all together as individuals may not be really able to perceive the moderators exclusively on their own. |
Thank you for your comment, nevertheless we have already written as premise that: “For each analysis, we ran process model number 1 in the macro developed by Hayes [74] and estimated the relationship between the predictor and the criterion at low (M - 1SD), medium and high (M + 1SD) levels of the supposed moderator. In each moderation model, self-esteem was co-varied out in order to control for the personal resources that may potentially influence the reactions to job demands. [Line 101] |
According to See et al.’s (2011) article, stigma is a multi-dimensional construct that was analyzed with first order and second order factor analyses. I would recommend multi-dimensional variables to be analyzed accordingly as well. The results may be easier to explain, especially as the current analysis form has significant and insignificant moderation findings. Another idea could be to test for a 4-way interaction. |
We thank the reviewer for their suggestion. Nevertheless, this study only represents a first testing of the mediation effect. The stigma scale has been used only in the sub dimensions as indicated by the author.
|
Was stigma measured by a Likert scale? The manuscript states it was a multiple-choice measure. |
In retrospect, we admit this was a typo: thanks for highlighting it, we fixed it in the revised version [Line 262] |
What is the purpose of self-esteem for the study? |
Self-esteem is part of variables assumed by the JD-R model as moderators between JD and outcomes. We have already considered it starting from the previous research (Ramaci et al., 2020) which did not refer to the JD-R model. It has been inserted as co-varied because we are convinced that it is essential to understand the possible contribution of personal factors already inherent in the theoretical reference model, but because the primary objective is to test the role of stigma in the present model. |
Regression tables should be included. |
Regression tables are synthetized in Table 2, and all the results are explained in the text |
The findings need to be further explained, especially with the moderation results |
Thanks for your advice. We have provided guidance on interpretations of moderation results in discussions [lines 461-478]. Anyway, given your suggestion, in order to fully give a possible explanation for the moderation results, we addedd the following lines at the end of the result section:
Referring to the JD-R model, the fact that the results showed that moderation by stigma is confirmed for negative outcomes and less for satisfaction, could confirm the difference in the processes expressed by the model that sees JDs, on the one hand, impacting on negative outcomes, while job resources associated with positive outcomes. The stigma would seem to act as a sort of booster in this relationship as if it were able to increase the effect of the perceptions of JD on outcomes only when both JD and stigma reach high levels. More complex to justify based on the JD-R model is the role of non-acceptance as a moderator, being able to impact the relationship between JD and both negative and positive outcomes: non-acceptance could have a general impact on outcomes but future research will certainly have to confirm the present results. |

Reviewer 3 Report
The paper deals with a very relevant and current topic such as the levels of quality of professional life at work experienced by workers on the frontlines during the pandemic, also evaluating the contribution of a purely social aspect such as the stigma towards clients/customers as possible COVID carriers.
The topics are discussed within a very strong, clear and defined theoretical framework (Job Demands-Resources model). The main hypothesis is that the social stigma can moderate the relationship between job demands and work outcomes: that is fatigue, burnout and satisfaction as components of professional quality of life
The paper presents potential to add knowledge to the topic. The paper is well written, the methodological approach is relevant and easy to understand. The results are clearly presented. The conclusions are in line with paper objectives and results.
Literature Review
The description of theoretical framework is clearly and punctually described. The hypotheses of study clearly derive from it. The bibliography and the theoretical references are complete and updated
Method
The method and research design are rigorous and clearly described. The information about procedure and sample description are adequate.
Results
The authors could consider whether or not to include a path diagram to facilitate understanding of the theoretical and moderation model
Discussion and conclusion
Results discussion and practical implications are consistent with theoretical premises and the aims/hypotheses of the research.
I would ask the authors to analyze the results of the moderation effects a little more deeply (specifically on the partial confirmation of Hp2c). For example, why is the moderation effect significant and involves the three dimensions of the stigma on the negative outcome (burnout and fatigue) and less dimensions in the positive one (satisfaction)? Why is non-acceptance significant for both of us, while the same cannot be said for discrimination or fear? The authors could answer these questions using previous literature or their reflections.
Author Response
I would ask the authors to analyze the results of the moderation effects a little more deeply (specifically on the partial confirmation of Hp2c). For example, why is the moderation effect significant and involves the three dimensions of the stigma on the negative outcome (burnout and fatigue) and less dimensions in the positive one (satisfaction)? Why is non-acceptance significant for both of us, while the same cannot be said for discrimination or fear? The authors could answer these questions using previous literature or their reflections. |
Thanks for your advice. We have provided guidance on interpretations of moderation results in discussions [lines 461-478]. Anyway, given your suggestion, in order to fully give a possible explanation for the moderation results, we addedd the following lines at the end of the result section:
Referring to the JD-R model, the fact that the results showed that moderation by stigma is confirmed for negative outcomes and less for satisfaction, could confirm the difference in the processes expressed by the model that sees JDs, on the one hand, impacting on negative outcomes, while job resources associated with positive outcomes. The stigma would seem to act as a sort of booster in this relationship as if it were able to increase the effect of the perceptions of JD on outcomes only when both JD and stigma reach high levels. More complex to justify based on the Jd-R model is the role of non-acceptance as a moderator, being able to impact the relationship between JD and both negative and positive outcomes: non-acceptance could have a general impact on outcomes but future research will certainly have to confirm the present results.
|

Reviewer 4 Report
A much needed study and good choice of variables to investigate. A sound theoretical base and solid arguments.
The structure of paragraphs need to be improved. There were many paragraphs that were one sentence - there should be a few sentences in each paragraph. This made reading the article more difficult to follow. It seemed like a series of important points, instead of one central point with supporting evidence (ideally each paragraph should have).
Line 56 - did you mean to say patients (not clients) as you were referring to previous study?
Some information on how the participants were selected in the method section is needed. We only learn it is a convenience sample in the limitations. Were they approached whilst at work? Online questionnaire or paper? Language?
Overall, a well executed study. The sample size was a little small considering the statistical analyses - was this checked?
Author Response
The structure of paragraphs need to be improved. There were many paragraphs that were one sentence - there should be a few sentences in each paragraph. This made reading the article more difficult to follow. It seemed like a series of important points, instead of one central point with supporting evidence (ideally each paragraph should have).
|
Thank you for your suggestions, we performed the structure of paragraphs. |
Line 56 - did you mean to say patients (not clients) as you were referring to previous study?
|
Dear reviewer in that case we generally referred to both clients and patients. In our opinion, if you agree, we would to keep clients because it includes both patients and users. |
Some information on how the participants were selected in the method section is needed. We only learn it is a convenience sample in the limitations. Were they approached whilst at work? Online questionnaire or paper? Language?
|
Dear reviewer, we thank you for your suggestion. We added some specifications to lines 224 “Overall, of the 413 total workers from the 5 involved supermarkets, 316 participated by filling out the paper questionnaire during working hours (response rate = 76%) and…” |
Overall, a well-executed study. The sample size was a little small considering the statistical analyses - was this checked?
|
Thank you for your suggestion. The sample is sufficient and given your useful suggestion we have indicated the procedure for verifying the appropriateness of the sample.
Following literature indications (Faul, Erdfelder, Lang & Buchner, 2007), a statistical power analysis was conducted through G*Power software to check for sample appropriateness; the minimum required sample emerged 270 (2 predictors including the interaction effect, effect size level = 0.15, α = 0.05, power requirement of 0.80). |

Round 2
Reviewer 2 Report
Although the manuscript has been improved, there are areas where my previous comments were not fully addressed in the manuscript.
First, I recommend an overarching framework to be mentioned in the introduction as it can clarify my previous comment about how the introduction can be strengthened.
Second, as I mentioned the lack of distinction between previous studies, this comment was not addressed at all. There are actually numerous ways to respond to this comment as it is a critical issue such as rehypothesizing the study.
Third, unfortunately, I am not too convinced about how the multiple moderators were analyzed. As mentioned before, Model 1 may not be appropriate, especially when comparing with extant studies that have analyzed multiple moderators.
Good luck with your future iterations and it was a pleasure reviewing your work.
Author Response
First, we recommend an overarching framework to be mentioned in the introduction as it can clarify my previous comment about how the introduction can be strengthened. |
Dear reviewer. The review work involved integrating the suggestions of 4 reviewers, and working on often conflicting comments and requests is complex, as you may already know. Some reviewers asked us to simplify the introduction because it was too long, while you asked us for additional changes. We’re really sorry we did not satisfy you with the explanation of the theoretical framework; we’re sorry to say that your comments were not fully clear to us, and somehow a little vague to manage a correct strengthening of the introduction. Anyway, again we added a long phrase in the introduction section, trying to clarify the theoretical aspects you mentioned, and a short specification in the study aims:
Lines 60 introduction (lines 60-66) From a theoretical point of view, there is also a substantial discrepancy in the conceptualization of the stigma at work (especially in an active form, i.e., of workers towards customers) within organizational models that account for the effects on the well-being of workers. According to Major & O'Brien (2005), stigma has an impact on the health of workers because it can be considered a sort of social demand or factor that reflects intergroup relationships at work, and should be considered within models such as the JD-R model to somehow go beyond or overcome the simple evaluation of working conditions.
Lines 183 study aims …. consequent to a social demand that reflects intergroups relationships at work (workers vs customers).
We really hope that these modifications are ok for you. If we didn’t manage to understand exactly what do you mean, please, we ask you to clarify exactly where (lines) and which content are missing from the introduction. Thank you for your patient.
|
Second, the lack of distinction between previous studies, this comment was not addressed at all. There are actually numerous ways to respond to this comment as it is a critical issue such as rehypothesizing the study. |
Lines 51-59 explore the previous study, while lines 174-182 justify the introduction of the JD-R model as a framework for the present research. In the discussion, lines 458-465, and lines 466-468, explore the original elements of the paper, the contribution to the theory, and how it differs from previous studies. Put simply, the research has two differentiators: 1) the sample is not of first-line health workers, but front-line workers. 2) the model, compared to the previous research, has been inserted within the theoretical framework of the JD-R model, and the model has been tested. Maybe you’re asking for a classical section or phrase in which we clearly state the originality of the present research?
We addedd the following phrase in the discussion section [lines 497-504]
Much research to date has not dealt with worker stigma towards customers and users, and its effect on work outcomes, focusing mainly on self-perceived stigma; equally few studies have attempted to integrate this construct within the JD-R model, proposing an interpretation of the stigma as a personal threat expression of a work-related social demand, and inserting it in the JD-R model as a moderator. In addition, the present research contributes to knowledge relating to the mechanisms of stigma in the pandemic context, in the context of workers exposed on the front line, and by extending the knowledge derived so far to non-health workers. |
